# Exploring the Relationship among Predictability, Prediction Accuracy and Data Frequency of Financial Time Series

**DOI:** 10.3390/e22121381

**Published:** 2020-12-06

**Authors:** Shuqi Li, Aijing Lin

**Affiliations:** School of Science, Beijing Jiaotong University, Beijing 100044, China; 18121635@bjtu.edu.cn

**Keywords:** entropy rate, predictability, entropy difference, financial time series

## Abstract

In this paper, we aim to reveal the connection between the predictability and prediction accuracy of stock closing price changes with different data frequencies. To find out whether data frequency will affect its predictability, a new information-theoretic estimator $P_{lz}$, which is derived from the Lempel–Ziv entropy, is proposed here to quantify the predictability of five-minute and daily price changes of the SSE 50 index from the Chinese stock market. Furthermore, the prediction method EEMD-FFH we proposed previously was applied to evaluate whether financial data with higher sampling frequency leads to higher prediction accuracy. It turns out that intraday five-minute data are more predictable and also have higher prediction accuracy than daily data, suggesting that the data frequency of stock returns affects its predictability and prediction accuracy, and that higher frequency data have higher predictability and higher prediction accuracy. We also perform linear regression for the two frequency data sets; the results show that predictability and prediction accuracy are positive related.

## 1. Introduction

As the most essential task of financial market analysis, price analysis has been paid more and more attention, even though the support for the strong version of the efficient market hypothesis (EMH) [1,2,3] has decreased since the 1980s [4,5]. If the EMH is of some relevance to reality, then a market would be very unpredictable due to the possibility for investors to digest any new information instantly [6,7]. However, new evidence challenges the EMH with many empirical facts from observations, e.g., the leptokurtosis and fat tail of the non-Gaussian distribution, especially the fractal market hypothesis (FMH) [8,9]. In addition, Beben and Orlowski [10] and Di Matteo et al. [11,12,13] found that emerging markets were likely to have a stronger degree of memory than developed markets, suggesting that the emerging markets had a larger possibility of being predicted.

Traditionally, econometricians and econophysicists are more interested in predictability of price changes in principle and in practice. The notion of predictability of the time series can be explained by the memory effects of the past values. Using entropy to measure the degree of randomness and the predictability of a series has been a topic for a long time; it goes back almost to the very beginning of the development of communication and information theory.

In this paper, we propose a new information-theoretic predictability estimator $P_{lz}$ for financial time series, which is derived from the Lempel–Ziv estimator [14,15,16]. The $P_{lz}$ quantifies the contributions of the past values by reducing the uncertainty of the forthcoming values in time series. Then we use the prediction method EEMD-FFH [17] to find some connections between the predictability and prediction accuracy of financial time series.

The paper is organized as follows. In the Methodology section, we introduce two predictability estimators which measure the magnitude of predictability of time series, $D_{norm}$ [18] and $P_{lz}$ proposed in this paper, and a prediction algorithm EEMD-FFH. In the Numerical Simulation section, we apply these two estimators to the artificial simulation numerical analysis, the logistic map. The Financial Time Series Analysis and Prediction section is the most important part in this paper, which draws two important results. Finally, the Conclusion section is devoted to the summing-up and future studies.

## 2. Methodology

### 2.1. Entropy Rate

Entropy rate can be used to estimate the predictability of time series. It is a term derivative of the entropy, which measures the uncertainty of a random process in theory of information. Let $x=\left\{x_{t}\right\},t=1,2,\ldots ,T$ be a time series realization. The Shannon entropy [19] of a random variable at time *t* is defined as:$$H\left[x_{t}\right]=-\sum_{x_{t}\in \Theta }p\left(x_{t}\right)\log_{2}p\left(x_{t}\right)$$
where $p\left(x_{t}\right)$ is the probability distribution of $x_{t}$ and Θ is sample space. Shannon also introduced the entropy rate, which generalizes the notion of entropy. For a stochastic process $X=\left\{X_{t}\right\},t=1,2,\ldots ,T$, the entropy rate is given by
$$H\left(X\right)=\lim_{T\to \infty }\frac{1}{T}H\left(X_{1},X_{2},\ldots ,X_{T}\right)$$
The right side can be interpreted as the average entropy of each random variable in the stochastic process. If the process satisfies the stationarity condition, the entropy rate can also be expressed as a conditional entropy rate
$$H\left(X\right)=\lim_{T\to \infty }H\left(\left.X_{T}\right|X_{1},X_{2},\ldots ,X_{T-1}\right)$$
It denotes the uncertainty in a quantity at time *T* having observed the complete history up to that point.

### 2.2. Entropy Rate Estimation

Entropy rate estimation has been paid more and more attention over the last 10 years due to the fact that the real entropy is known in very few isolated applications, one of the main reasons being the crucial practical importance of information-theoretic techniques in neurosciences. Entropy rate estimators can be classified into two categories [20]:**i.** The “plug in” (also called maximum-likelihood) technique and its modifications. The main principle of these methods is to compute the empirical frequencies of different patterns in the data, and then calculate the entropy of the empirical distribution. Due to the cost of calculation and limits on the data size, the “plug in” method cannot reveal the signal with long term time dependency.**ii.** Estimators based on data compression methods, such as Lempel–Ziv (LZ) [14,15,16] and context-tree weighting (CTW) [21,22]. This kind of approach is used to speed up the convergence and improve the performance in capturing long term time dependency.

In this study, we use two estimators which fall into the above two categories, entropy difference [18] belonging to the “plug in” class, and the new estimator we propose $P_{lz}$ belonging to the Lempel–Ziv class).

#### 2.2.1. $D_{norm}$ Predictability Estimator

Consider a time series $X=\left\{x_{t}\right\},t=1,2,\ldots ,T$. The entropy rate at time *t* for a stationarity process is defined as $H\left[x_{t}\left|x_{1},x_{2},\ldots ,x_{t-1}\right.\right]$. We assume that the underlying system can be approximated by a *p*-order Markov process. Then the value of the current moment is only related to the previous *p* moments. Hence, we can simplify the entropy rate:$$H\left[x_{t}\left|x_{1},x_{2},\ldots ,x_{t-1}\right.\right]=H\left[x_{t}\left|x_{t-p},x_{t-p+1},\ldots ,x_{t-1}\right.\right]$$
$$\equiv H\left[x_{t}\left|x_{t-1}^{\left(p\right)}\right.\right]=\sum_{x_{t},x_{t-1}^{\left(p\right)}\in \Theta }p\left(x_{t},x_{t-1}^{\left(p\right)}\right)\log_{2}\frac{p\left(x_{t},x_{t-1}^{\left(p\right)}\right)}{p\left(x_{t-1}^{\left(\pi \right)}\right)}$$

After we consider the past values, the uncertainty of the time series will not increase; therefore, $H\left[x_{t}\left|x_{t-1}^{\left(p\right)}\right.\right]\leq H\left[x_{t}\right]$. Now we can define the entropy difference (ED) as *D*, which is the difference between the entropy and entropy rate and is non-negative.
$$D=H\left[x_{t}\right]-H\left[x_{t}\left|x_{t-1}^{\left(p\right)}\right.\right]$$

The right side can be interpreted as the contributions of the past values to reduce the uncertainty at time *t*. If the underlying process is a random walk, then D=0. That is to say, the past values provide no information for current time. $0<D\leq H\left[x_{t}\right]$ indicates that the process has time autocorrelation; thus, the past values can help to improve the predictability at time *t*. Due to the lower bound and upper bound being certain of *D*: $0<D\leq H\left[x_{t}\right]$, we can normalize it to interval $\left[0,1\right]$.
$$D_{norm}=\frac{H\left[x_{t}\right]-H\left[x_{t}\left|x_{t-1}^{\left(p\right)}\right.\right]}{H\left[x_{t}\right]}=1-\frac{H\left[x_{t}\left|x_{t-1}^{\left(p\right)}\right.\right]}{H\left[x_{t}\right]},\;0\leq D_{norm}\leq 1$$
$D_{norm}$ measures the predictability of time series. When $D_{norm}$ tends to 0, the time series is unpredictable. If $D_{norm}$ is approximately 1, $H\left[x_{t}\left|x_{t-1}^{\left(p\right)}\right.\right]\approx 1$, the time series can be predicted completely at time *t*.

We proceeded with three numerical simulations to apply $D_{norm}$ to different time series respectively. The data size is 10,000 points. The results are like those in Table 1. The first row is the entropy and $D_{norm}$ of a deterministic time series 1,1,…,1. The simulation results are consistent with our intuitive understanding, namely, the uncertainty is 0 or the predictability is 1. The results of repeated pattern indicates that for a repeat pattern time series, the entropy is 1 up to the upper bound, and the underlying time series can be predicted totally when past values were considered. For the pure random series, the predictability equals to 0. That is, we cannot predict a pure random time series even we consider its past values. The result is dependent on three factors: sample size, the efficiency of the estimator and the quality of the random generator. Hence, it is easy to understand why the entropy is 0.9997 not equal to 1.

#### 2.2.2. $P_{lz}$ Predictability Estimator

After Kolmogorov defined the complexity as the size of the minimum binary code that produces this time series in 1965 [23], complexity has been widely used to estimate entropy rate. Jacob Ziv and Abraham Lemple in 1977 designed a practical algorithm called Lempel–Ziv [16] to measure the complexity in the Kolmogorov sense, which also can identify the randomness of a time series. On this basis, there may entropy rate estimators were derived. One of them was created by Kontoyiannis in 1998 [24] (it will denoted as $H_{lz}$ in the better). $H_{lz}$ was widely used, and proved to have good statistical properties and better practical performance than other Lempel–Ziv estimators [24].

Consider a time series $X=\left\{x_{t}\right\},t=1,2,\ldots ,T$. The $H_{lz}$ estimator is defined as:$$H_{lz}={\left(\frac{1}{T}\sum_{i=1}^{T}L_{i}\right)}^{-1}\log_{2}T$$
where *T* is the size of the underlying time series and $L_{i}$ is the length of the shortest substring starting from time *i* that does not appear as a contiguous substring in the prior values.

It has been proved that $H_{lz}$ converges to the entropy rate with probability one as *T* approaches infinity for a stationary ergodic Markov process.

We calculate the estimator $H_{lz}$ values of some simulated time series, which were generated at random from the alphabet {1,2,3,4,5,6,7,8} with different data sizes *T*. The theoretical entropy is equal to $-\sum_{i=1}^{8}\left(1/8\right)\log_{2}\left(1/8\right)=3$. As shown in the left part of Figure 1, $H_{lz}$ converges to this theoretical value as data size *T* tends to infinity.

In our study, we propose a new predictability estimator $P_{lz}$, which is derived from the estimator $H_{lz}$. $P_{lz}$ is defined as:$$P_{lz}=1-H_{lz}/\hat{S}$$
where $\hat{S}=\log_{2}S$ and S is the number of distinct states of the symbolized data. The estimator $H_{lz}$ is normalized into interval $\left[0,1\right]$. We will introduce the detailed discretized method and its necessity in Section 3.

### 2.3. EEMD-FFH Prediction Algorithm

In this paper we will use a particular method EEMD-FFH [17] to find out whether the predictability of time series is related to the prediction accuracy of a particular algorithm.

The EEMD method is used to decompose a time series into a series of intrinsic mode functions (IMFs) and one residue. It has been widely used in various industries [25,26], and was proposed by Huang et al. [27]. Based on the EEMD model, there is a hybrid prediction model called EEMD-FFH [17,28] that integrates MKNN (for predicting high frequency IMFs), ARIMA (for predicting low frequency IMFs) and quadratic regression (for residue wave) models.

The operation steps of EEMD-FFH are as follows:**Step 1.** Decompose the time series X(t) via EEMD
$$X\left(t\right)=\sum_{i=1}^{n}c_{i}+r_{n}.$$**Step 2.** Use different models to predict IMFs of different frequencies
$$IMF_{i}⟹Result_{i}$$
$$r_{n}⟹Result_{r}$$**Step 3.** Sum up the results to get the prediction value
$$\hat{X}=\sum_{i=1}^{n}Result_{i}+Result_{r}$$

In the experimental section of this article, we use this algorithm to predict daily financial data and five-minute high-frequency financial data.

## 3. Numerical Simulation

In this section, we consider a nonlinear system, the logistic map, to test the two predictability estimators as mentioned above. Chaos in dynamical systems has been investigated over a long period of time. With the advent of fast computers, the numerical investigations on chaos have increased considerably over the last two decades, and by now, a lot is known about chaotic systems. One of the simplest and most transparent system exhibiting order to chaos transition is the logistic map [29]. The logistic map is a discrete dynamical system defined by
$$x_{t+1}=px_{t}\left(1-x_{t}\right)$$
with $0\leq x_{t}\leq 1$. Thus, given an initial value (seed) $x_{0}$, the series *x* is generated. Here the subscript *t* plays the role of discrete time. The behavior of the series as a function of the parameter *p* is interesting. A thorough investigation of logistic map has already been done [29]. Here, without going into detailed discussion, we simply note that

The logistic map has x=0 and x=(p−1)/p as fixed points. That is, if $x_{i}=0$ or $x_{t}=\left(p-1\right)/p$, then $x_{t+1}=x_{t}$.For p<1, x=0 is an attractive (stable) fixed point. That is, for any value of the seed $x_{0}$ between 0 and 1, $x_{t}$ approaches 0 exponentially.For 1≤p≤3, x=(p−1)/p is an attractive fixed point.For 3<p<3.56995, the logistic map shows interesting behavior such as repeated period doubling, appearance of odd periods, etc.Most values of *p* beyond 3.56995 exhibit chaotic behavior.

Here, we set p=3.7 and let the data length $N=10^{5}$. The initial value of $x_{0}$ is set to 0.5.

As only one equation is described in the logistic map, $x_{t}$ changes no information with other variables. We added Gaussian white noises to the original time series $x_{t}$ with different strengths to obtain a composite time series, $y_{t}=x_{t}+\lambda \epsilon_{t}$. $\epsilon_{t}$ is the Gaussian white noise (with zero mean and unit variance). λ≥0 is a parameter that tunes the strength of noise. $x_{t}$ is the real signal corrupted by the external noise $\epsilon_{t}$, and λ determines the signal–noise ratio. The larger the λ, the smaller the signal–noise ratio.

We used k-means clustering to discretize the data with added Gaussian white noises into *b* distinct clusters. *b* is a pre-defined parameter that determines the number of clusters. In Figure 2, we show the values of $D_{norm}$ and $P_{lz}$ on different bins b=5,6,…,14, with the noise strength parameter λ from 0.01 to 0.1 with a step of 0.01. The result indicates that the predictability of the time series decreased with increasing λ, as the signal–noise ratio became lower. $D_{norm}$ and $P_{lz}$ reach values close to 0.1 when λ=0.1, so it is hard to predict the composite time series when we add more Gaussian white noise into it. Moreover, the predictability of a logistic map has no obvious relationship with the number of bins. This result is consistent with [30], which found the choice of bins is largely irrelevant to the estimation results. Here, the parameter *b* for the k-means clustering was 10. This experimental setup has been proven to be very efficient at revealing the randomness of the original data [31].

From the above numerical simulations, we were able to conclude that the two estimators have good performances in estimating the randomness or predictability of the system (the predictability of the time series decreases with increasing λ), so we carried out the following real financial data experiments.

## 4. Financial Time Series Analysis and Prediction

### 4.1. Data and Stock Selection

In order to assess whether the five-minute high-frequency financial data or daily financial data has a larger possibility of being predicted, we estimated the entropy rate of close price of the stocks that make up the SSE (Shanghai Securities Exchange) 50 index. This index is based on scientific and objective methods to select the most representative 50 stocks with the large scale and good liquidity in the Shanghai stock market, able to form sample stocks, so as to comprehensively reflect the overall situation of a group of leading enterprises with the most market influence in Shanghai’s stock market. The data have been found at URL http://www.10jqka.com.cn/ and were up to date as of the 13 January 2019, going 10 years back. Only complete records, i.e., five-minute and daily data with valid values for both 50 stocks, were admitted; invalid values were filtered out. In reality, non adjacent data may become adjacent data because of this procedure, but the relatively small number of invalid values compared to the valid values prevents a statistically significant impact [32]. The original close data of SSE 50 stocks cannot reasonably be assumed as stationary, a property for a time series yet essential for the validity of the forthcoming analysis. A classical solution to solve this problem is to define some new variables which can be considered stationary or at least asymptotically stationary [33]. The usual transforms for raw time series $X=\left\{x_{t}\right\},t=1,2,\ldots ,T$ are as follows:$$\begin{array}{cc} Increment\;\delta x_{\tau }\left(t\right) & :=x(t+\tau )-x(t)\\ Return\;r_{\tau }\left(t\right) & :=\delta x_{\tau }\left(t\right)/x\left(t\right)\\ Log-Return\;s_{\tau }\left(t\right) & :=\ln[x(t+\tau )]-\ln[x(t)] \end{array}$$
The choice of the variable does not affect the outcome of the present work; in fact, in the high-frequency regime they are approximately identical or proportional to each other [33]. We will use the log-returns in the forthcoming analysis. The usual quantity employed to characterize the fluctuation in financial data is the so called volatility, here defined as
$${\mathrm{vol}}_{\Delta }\left(t\right):=\frac{1}{\Delta }\sum_{i=1}^{\Delta }\left|s_{\tau }\left(t+i\right)\right|$$
where the parameter Δ refers to the chosen length of the time-window and τ (in our cases always τ=1day&5min) denotes the basic time scale. The average values of the whole 50 stocks log-returns are ${\hat{s}}_{1\mathrm{day}}\left(t\right)≃\pm 1\times 10^{-4}$ and ${\hat{s}}_{5\min}\left(t\right)≃\pm 1\times 10^{-5}$, while the absolute log-returns, also interpretable as estimates of the 5min and daily volatility, have mean values of $v\hat{o}l_{1\mathrm{day}}\left(t\right)≃v\hat{o}l_{5\min}\left(t\right)≃6\times 10^{-4}$. However, as is widely known, the strength of fluctuations in financial data is subject to long-term correlated oscillations. Still, in concordance with other authors [33], we assume a sufficiently long financial time series to be asymptotically stationary, i.e., leading to relevant results for the long-term statistical properties of the analyzed data. The distributions of the Shannon entropy for daily data and five-minute high-frequency are shown in Figure 3. μ and σ represent mean and standard deviation respectively.

The results show that the entropy of 5 min closing prices is lower than that of daily closing prices. This is not very surprising, since high entropy has been observed even for larger time scales [34]. We considered 20 stocks, which included the five highest entropy stocks and the five lowest entropy stocks of the daily data and the same choice for the five-minute high-frequency financial data. In order to eliminate the influence of multi-scale on entropy calculation and stock selection, we used a coarse-graining algorithm by amplification in different proportions (range from 0 to 20). Then we calculated the average value and the median value for the coarse-graining dataset to choose high entropy stocks and low entropy stocks. The stocks selected by average value and median value were totally identical. After removing overlapping stocks, we obtained 20 stocks, and the detailed calculation results are shown in Table 2. These 20 stocks are considered in the experiments in the next sections.

To give the evidence that the raw time series are not random, we compared the entropy of the raw time series with the entropy of randomly shuffled variants of the original data, which is also called surrogate testing. With such a preprocessing, all potential correlations in the original time series were destroyed; 100 shuffled time series for each raw time series (before the homogeneous partitioning) were generated and their average entropy was measured. The distributions of shuffled data are different to those in the original time series, and the average entropy is much larger, as can be seen in Figure 4. This provides evidence that there are temporal dependencies in the data we analyzed, and it makes sense for us to calculate their degrees of predictability.

### 4.2. Estimating the Predictability of Different Frequency Time Series Based on $D_{norm}$ and $P_{lz}$

In this section, we use the 20 stocks already obtained in Table 2 to calculate the predictability of daily data and five-minute high-frequency data respectively, based on $D_{norm}$ and $P_{lz}$. The main question asked in this paper is whether daily price changes are more or less predictable than intraday (five-minute high-frequency) price changes. The reason why we use these two predictability estimators is to make the experiment more credible, and the two estimators are not compared in this paper.

We divide those 20 stocks into four groups, every group including five stocks, as shown in Figure 5, which we obtained in Section 4.1. Then for every part we calculate the predictability of daily data and five-minute high-frequency data respectively, based on $D_{norm}$ and $P_{lz}$. In Figure 6, the left panels show the predictability of every group based on $D_{norm}$, and the right panels show the predictability of every part based on $P_{lz}$.

Surprisingly, for every stock of every group the predictability value of five-minute high-frequency data is obviously much larger than daily data, which means that five-minute high-frequency price changes are more predictable than daily price changes. The experimental results strongly suggest that the predictability of time series is related to the frequency of the data itself. From this conclusion we raise another question: whether the predictability of time series is related to the prediction accuracy of a particular algorithm. In the next section we will focus on this question.

### 4.3. Comparing the Prediction Accuracies of Different Frequency Time Series Based on EEMD-FFH

In last section, the experimental results strongly suggest that five-minute high-frequency price changes are more predictable than daily price changes. Then, is it possible that high frequency time series, which are more predictable, have higher prediction accuracy? In this section, we detail another experiment based on EEMD-FFH algorithm to explore this.

In order to assess the performance of EEMD-FFH for data at different frequencies, we use an indicator, root mean squared error (RMSE).
$$RMSE=\sqrt{\frac{1}{n}\sum_{t=1}^{n}{\left(x_{t}-{\hat{x}}_{t}\right)}^{2}}$$
where $x_{t}$ represents the raw data; ${\hat{x}}_{t}$ represents the prediction value; *n* is the number of prediction points; smaller RMSE means higher accuracy.

Table 3 tabulates the average RMSE of five stocks in every group. The last 200 points for every time series have been predicted and the set containing these 200 points was our testing set. For every one of them we can see that the five-minute high-frequency financial data have higher prediction accuracy than daily data. We also show RMSE of every group in Figure 7, where the obvious difference is more intuitive.

To show the relationship between the predictability and prediction accuracy, we conducted correlation analysis. We calculated the Pearson correlation coefficient and Spearman correlation coefficient between the predictability and RMSE in different frequency, as shown in the following Table 4. These results reveal that the connection between these two concepts exists, and they are negatively correlated.

To explore the relationship further, we also performed linear regression for the two frequency data sets (every set includes 20 stocks of 4 groups). In this linear regression model, the value of predictability is an independent variable and the value of RMSE is the dependent variable. The scatter plot and fitted lines are shown in Figure 8. Every regression line shows that the predictability and RMSE are negatively correlated. RMSE is root mean squared error; high RMSE denotes low prediction accuracy—that is to say the predictability and prediction accuracy are positively related.

In statistics, the predictability of time series belongs to the category of time series analysis, which is different from the prediction accuracy based on a forecasting method. In this experiment, our goal was to see if the two were related. Surprisingly, the analysis results indicate that predictability fits prediction accuracy perfectly, and we found that the five-minute high-frequency financial data have higher predictability and prediction accuracy than daily data.

## 5. Conclusions

In this paper, we introduced a new information-theoretic predictability estimator $P_{lz}$ for financial time series, which was derived from the Lempel–Ziv estimator. The $P_{lz}$ quantifies the contributions of the past values by reducing the uncertainty of the forthcoming values in the time series. We limited ourselves to the stocks constituting SSE 50 index because they are primary components of the Chinese market, to do an experiment to explore whether data’s frequency would effect its predictability. The results strongly suggest that five-minute high-frequency price changes are more predictable than daily price changes. Additionally, we used the prediction method EEMD-FFH to find some connections between the predictability and prediction accuracy. Here, the empirical evidence suggests that there is a strong positive relationship between these two concepts—this is, higher frequency data have higher predictability and higher prediction accuracy.

Further studies should be performed to confirm whether these results are robust and valid for other stock markets as well. Another important study is to find whether different prediction methods will change the result that a strong positive relationship exists between predictability and prediction accuracy for different frequency financial data.

## Figures and Tables

**Figure 1 entropy-22-01381-f001:**
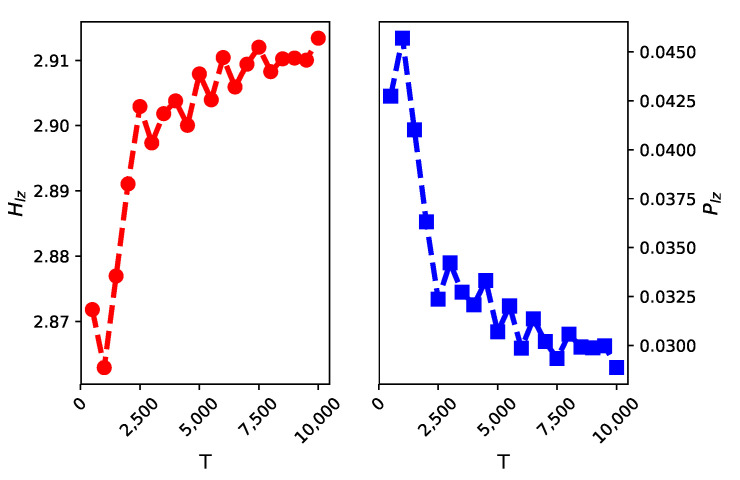
We calculate the estimator values ($H_{lz}$ on the left side and the right for $P_{lz}$) for different data volumes T={500,1000,…,10,000}. The results indicate that $H_{lz}$ converges to the theoretical entropy 3 and $P_{lz}$ converges to 0 as *T* grows.

**Figure 2 entropy-22-01381-f002:**
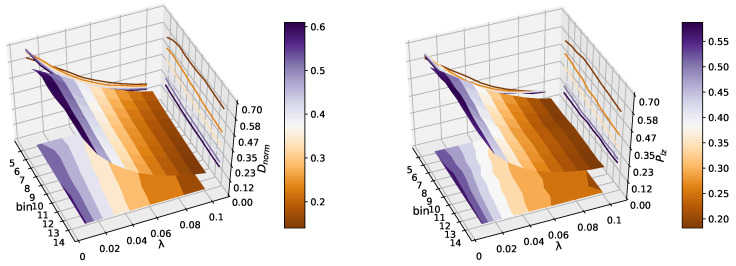
This figure shows that the change of predictability of the logistic map with increasing λ and bin. $D_{norm}$ and $P_{lz}$ reach values close to 0.1 when λ=0.1, so it is hard to predict the composite time series when we add more Gaussian white noises into it. Moreover, the predictability of logistic map has no obvious relationship with the number of bins, which we can see in the projection of the 3D image. There is not much difference between the two estimators, eliminating the effect of different statistics on the same systems.

**Figure 3 entropy-22-01381-f003:**
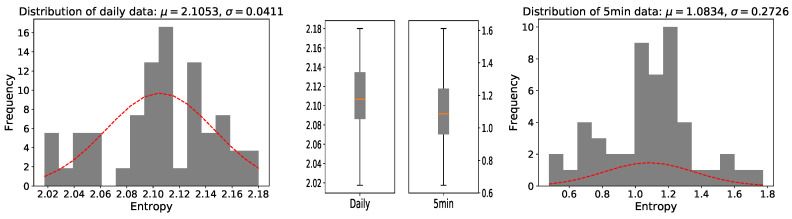
Distribution of the entropy of the stocks that make up the SSE 50 index (the left one shows log-returns of consecutive daily closing prices, while the right one shows log-returns of consecutive 5min closing prices). In every histogram, a normal distribution with the same mean and standard deviation is plotted. Discrete cases of two distributions in the two box-plots are shown.

**Figure 4 entropy-22-01381-f004:**
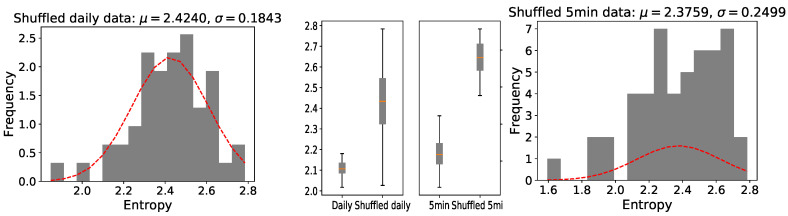
Distribution of the shuffled data entropy of the stocks that make up the SSE 50 index (the left one shows log-returns of shuffled daily closing prices while the right one shows log-returns of shuffled 5min closing prices). In every histogram, a normal distribution with the same mean and standard deviation is plotted. The entropy of surrogate time series is much larger than that of the raw data in the middle box-plots.

**Figure 5 entropy-22-01381-f005:**
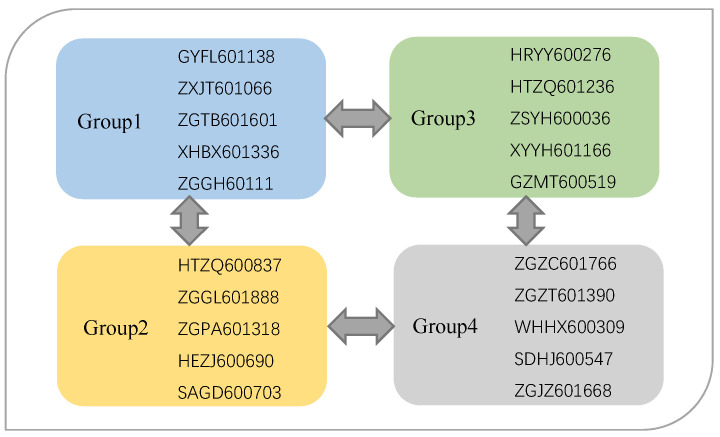
The 20 selected stocks we obtained in Section 4.1 are assigned to four groups to test whether daily price changes are more or less predictable than intraday (five-minute high-frequency) price changes.

**Figure 6 entropy-22-01381-f006:**
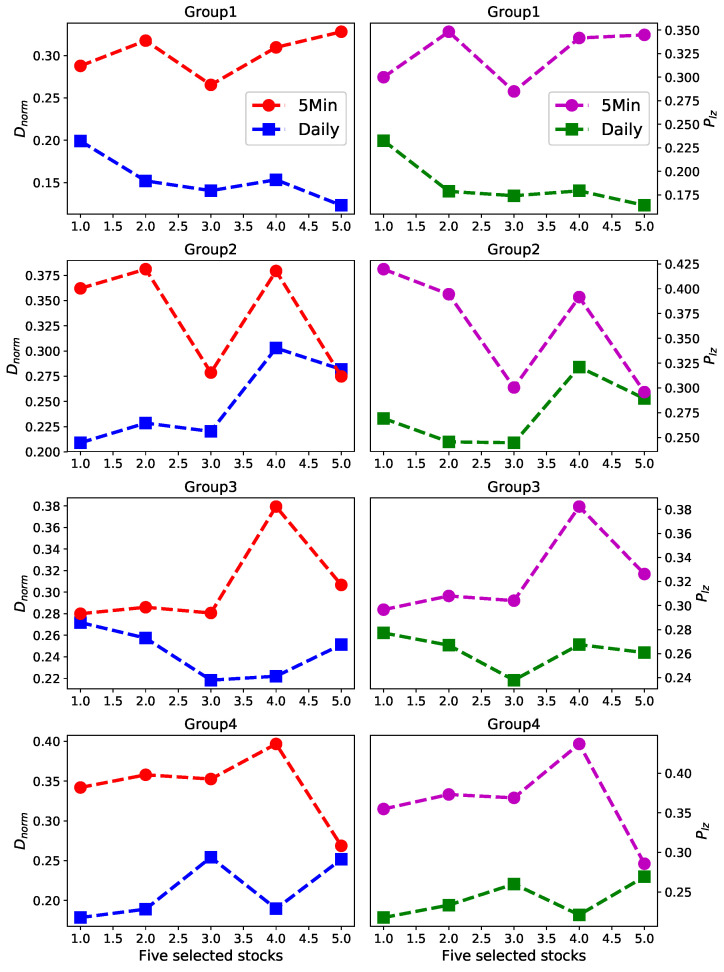
The predictability of daily data and five-minute high-frequency data respectively based on predictability estimators $D_{norm}$ and $P_{lz}$. Left panels show the results $D_{norm}$ for group1–4, respectively. Right panels show the results $P_{lz}$ for group1–4, respectively.

**Figure 7 entropy-22-01381-f007:**
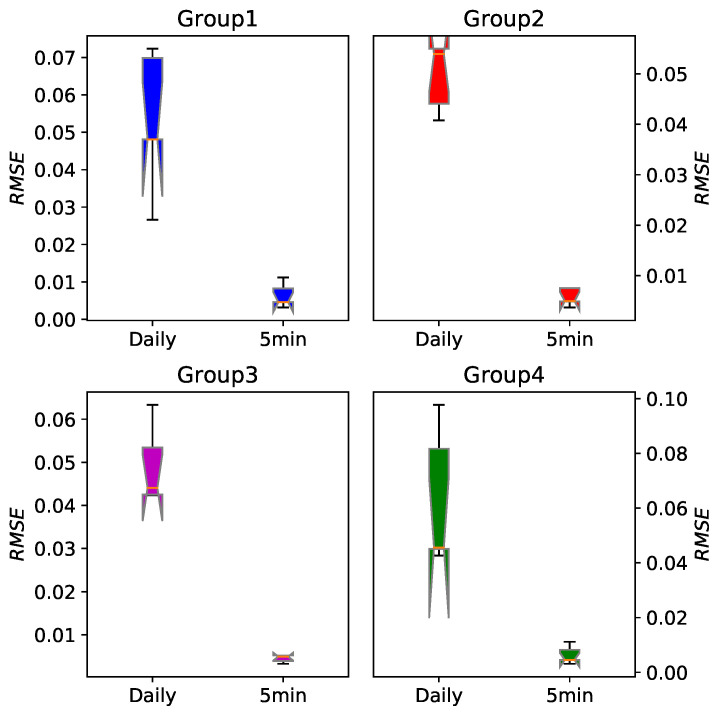
The boxplot of prediction error RMSE for the daily data and five-minute data based on the EEMD-FFH algorithm. For every group we can see that the five-minute high-frequency financial data have significantly higher prediction accuracy.

**Figure 8 entropy-22-01381-f008:**
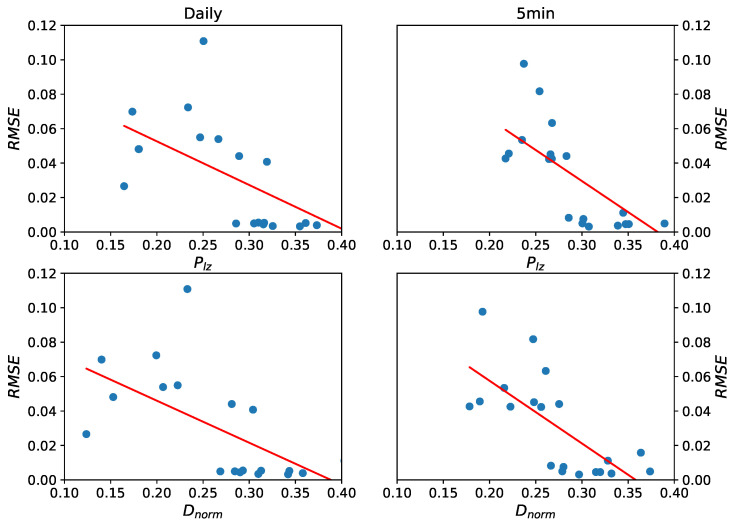
The scatter plot and regression fitted lines of daily data (**left panels**) and five-minute high-frequency data (**right panels**) respectively, based on predictability estimators $D_{norm}$ and $P_{lz}$. Every regression line shows that the predictability and RMSE are negatively correlated—that is to say the predictability and prediction accuracy are positive related.

**Table 1 entropy-22-01381-t001:** The entropy and $D_{norm}$ of three time series.

Time Series (N = 10,000)	Entropy	$D_{norm}$
All 1: 1, 1, …, 1	0	1
Repeated pattern (0, 1): 0, 1, 0, 1, …, 0, 1	1	1
Generate random integer from 1, 2	0.9997	0

**Table 2 entropy-22-01381-t002:** The selected stocks by calculating the average value and the median value for the coarse-graining dataset.

Data Type	Daily				5 min			
Method	Average		Median		Average		Median	
High entropy	**GYFL601138**	2.416	**GYFL601138**	2.507	**HRYY600276**	1.592	**HRYY600276**	1.772
	**ZXJT601066**	2.385	**ZXJT601066**	2.346	**HTZQ601236**	1.454	**HTZQ601236**	1.611
	**ZGTB601601**	2.193	**ZGTB601601**	2.261	**ZSYH600036**	1.452	**ZSYH600036**	1.598
	**XHBX601336**	2.119	**ZGGH601111**	2.152	**XYYH601166**	1.417	**GZMT600519**	1.536
	**ZGGH601111**	2.118	**XHBX601336**	2.148	**GZMT600519**	1.372	**XYYH601166**	1.436
Low entropy	**HTZQ600837**	1.310	**ZGPA601318**	1.380	**ZGZC601766**	0.661	**ZGZT601390**	0.666
	**ZGGL601888**	1.299	**HTZQ600837**	1.343	**ZGZT601390**	0.660	**WHHX600309**	0.658
	**ZGPA601318**	1.276	**ZGGL601888**	1.295	**WHHX600309**	0.605	**ZGZC601766**	0.647
	**HEZJ600690**	1.260	**HEZJ600690**	1.215	**SDHJ600547**	0.579	**SDHJ600547**	0.542
	**SAGD600703**	0.930	**SAGD600703**	0.979	**ZGJZ601668**	0.524	**ZGJZ601668**	0.478

**Table 3 entropy-22-01381-t003:** RMSE of the two different data frequencies based on the EEMD-MKNN algorithm.

Group	Daily Data	5 min Data
1	0.0530	0.0063
2	0.0609	0.0073
3	0.0491	0.0047
4	0.0625	0.0056

**Table 4 entropy-22-01381-t004:** Correlation coefficient between the predictability and RMSE.

	Daily Data		5 min Data	
	Pearson	Spearman	Pearson	Spearman
$P_{lz}$	−0.6081	−0.7018	−0.7022	−0.7649
$D_{norm}$	−0.6223	−0.6837	−0.7129	−0.7589

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
