# Peer review of "Exploring the Relationship among Predictability, Prediction Accuracy and Data Frequency of Financial Time Series"

_entropy, 2020, doi:10.3390/e22121381_

Round 1
Reviewer 1 Report
See the attached file

Reviewer 2 Report
see the attachment

Round 2
Reviewer 1 Report
See the attached file.

Author Response
Please find the responses in the attachment.

Reviewer 2 Report
In the past report, I concerned the estimators derived due to (1) entropy measure that determine the predictability and (2) some measures of prediction accuracy. And also states the differences between these two types of estimators. The authors only mention the estimators using "Plug in" method and estimators based on data compression algorithm. I am not sure the authors answer the question properly. Besides, I have mentioned that "It is not clear that the connection of the predictability and prediction accuracy that is claimed to be revealed. This could be shown numerically or theoretically." However the authors suggest that that would happen in future research. So I am not sure how this article "aims to reveal the connection between the predictability and prediction1accuracy of stock closing price changes with different data frequencies" written first sentence in the abstract.
Author Response

(The authors gave the same response as above.)

Round 3
Reviewer 1 Report
See the attached file.

Author Response

(The authors gave the same response as above.)

Reviewer 2 Report
In the abstract, last sentence says that "It turns out that intraday five-minute data are more predictable and also have higher prediction accuracy than daily data, suggesting that data frequency of stock returns affects its predictability and prediction accuracy and that higher frequency data have higher predictability and higher prediction accuracy." that is also mentioned in the conclusion. However, in the newly added paragraph in page 11, it suggests that predictability and RMSE are negatively correlated. Here I guess RMSE measures the prediction accuracy. So there is a contradiction of the material and the authors might want to explain that.
Author Response

(The authors gave the same response as above.)
